# Effects of *Sophora alopecuroides* in a High-Concentrate Diet on the Liver Immunity and Antioxidant Function of Lambs According to Transcriptome Analysis

**DOI:** 10.3390/ani14020182

**Published:** 2024-01-05

**Authors:** Yawen An, Hairong Wang, Aiwu Gao, Shufang Li, Jinli Yang, Boyang Li, Henan Lu

**Affiliations:** 1College of Animal Science, Inner Mongolia Agricultural University, Hohhot 010018, China; 18648041184@163.com (Y.A.); lishufang0325@163.com (S.L.); yangjinli207@aliyun.com (J.Y.); 15513839921@163.com (B.L.); lhena1117@163.com (H.L.); 2Key Laboratory of Animal Nutrition, Animal Nutrition and Feed Science, Hohhot 010018, China; 3College of Food Science and Engineering, Inner Mongolia Agricultural University, Hohhot 010018, China; nmndgaw@126.com

**Keywords:** lamb, liver, immunization, antioxidant, LPS, inflammation, transcriptome

## Abstract

**Simple Summary:**

Prolonged feeding of excessive fermentable concentrates reduces rumen acidity and causes gram-negative bacteria to lyse and release lipopolysaccharides, triggering an inflammatory response in the body. As an important immune and metabolic organ in animals, the liver is susceptible to damage. Although the Chinese herb *Sophora alopecuroides* has antitumor, anti-inflammatory, antioxidant and other pharmacological effects, the mechanism of action of *Sophora alopecuroides* on the anti-inflammatory and antioxidant functions of the liver in lambs is still unclear. In this study, via liver function indices, liver inflammatory factor levels, antioxidant indices and transcriptome analysis, we demonstrated that *Sophora alopecuroides* could improve the immune and antioxidant functions of the liver in lambs under high-concentrate conditions and regulate the mechanism of liver injury in sheep through the expression of the relevant genes involved in the ECM–receptor interactions and the focal adhesion pathway. Elucidating the molecular mechanism of inflammatory responses is crucial for understanding liver dysfunction induced by high-concentrate diets.

**Abstract:**

The purpose of this study was to determine the effects of *Sophora alopecuroides* (SA) on liver function, liver inflammatory factor levels, antioxidant indexes and transcriptome in sheep. Twenty-four 3-month-old healthy Dumont hybrid lambs weighing 25.73 ± 2.17 kg were randomly divided into three groups: C1 (the control group), fed a concentrate-to-forage ratio of 50:50; H2 (the high-concentration group), fed a concentrate-to-forage ratio of 70:30; and S3 (the SA group), fed a concentrate-to-forage ratio of 70:30 + 0.1% SA. The results showed that the rumen pH values of the C1 and S3 groups were significant or significantly higher than that of the H2 group (*p* < 0.05 or *p* < 0.01). The serum ALT, AST and LDH activities and the LPS and LBP concentrations in the sheep serum and liver in the H2 group were significantly or extremely significantly higher than those in the C1 and S3 groups (*p* < 0.01), and the IL-10 content and SOD, GPX-P_X_ and T-AOC activities showed the opposite trend (*p* < 0.05 or *p* < 0.01). KEGG enrichment analysis showed that the differentially expressed genes were significantly enriched in the ECM–receptor interaction and focal adhesion pathways, which are closely related to immune and antioxidant functions (*p*-adjust < 0.1). In summary, SA could improve the immune and antioxidant functions of lamb livers under high-concentrate conditions and regulate the mechanism of damage on sheep livers, which is caused by high-concentrate diets and through the expression of related genes in the ECM/FAs pathway.

## 1. Introduction

In modern animal husbandry, to pursue high-yield meat and dairy products for greater profitability, ruminants are fed high-concentrate diets to meet the animals’ own high-performance needs. However, excessive fermentable concentrates lead to decreased rumen acidity in ruminants, and gram-negative bacteria are lysed and die to release LPS, which triggers an inflammatory response in the body [1]. LPS is a major component of the cell wall of gram-negative bacteria and is a major factor in inducing inflammatory damage in the body [2]. Thus, at low pH, the rumen epithelial barrier is disrupted and LPS may be transferred from the intestinal mucosa to the blood circulation where it can stimulate the release of proinflammatory cytokines [3,4]; additionally, the transfer of LPS from the rumen to the liver via the portal bloodstream triggers an inflammatory cascade response [5,6]. The liver is an important immune organ in animals. The liver is referred to as the body processing factory and is the main organ responsible for metabolism and detoxification [7]. Most toxins are decomposed, synthesized and detoxified in the liver, which is why these organisms are more susceptible to liver injury, fibrosis, cirrhosis, cancer and other liver diseases [8]. Therefore, elucidating the molecular mechanisms underlying the inflammatory response is critical for understanding high-concentration diet-induced liver dysfunction.

SA is a perennial herb that is produced in northwestern China and in several Asian countries in the arid desert grasslands [9], with antitumor, antibacterial, anti-inflammatory, antipyretic, analgesic and other pharmacological activities [10]. Previous pharmacological studies have shown that alkaloids are the main bioactive components in SA, followed by flavonoids, proteins, organic acids, amino acids and sugars [11]. Studies have confirmed that proinflammatory cytokines, including TNF-α, IL-6 and TGF-β, play key roles in the conversion of steatosis to steatohepatitis, which ultimately accelerates liver fibrosis [12,13]. The alkaloids in SA can inhibit the synthesis of proinflammatory cytokines, thus reducing the occurrence of hepatitis [14]. Wu et al. [15] reported that the total alkaloids (TAs) of SA can reduce the ALT, AST and MDA contents in the serum of mice and increase the SOD activity, indicating that TASA has a certain protective effect on alcohol-induced liver injury in mice, which may be related to the ability of TASA to alleviate oxidative stress. Previous studies conducted in our laboratory have shown that SA could alleviate the negative effects of high-concentrate conditions on lambs, and a feeding dose of 0.1% is appropriate [16,17]. However, the mechanism through which SA affects the immune and antioxidant functions of lamb livers under high feed conditions is still unclear. RNA-Seq provides important insights into genetic information and biological functions and can be used to explore the molecular mechanisms of animal phenotypes at the gene expression level, and in recent years, it has been widely used to reveal the biological functions of natural Chinese herbs in animals [18,19,20]. Thus, the aim of this study was to investigate the effects of SA on the immune and antioxidant functions of lamb livers under high-concentrate conditions and to elucidate the molecular mechanism through which SA alleviates the inflammatory response of lamb livers under high-concentrate conditions via RNA-Seq to provide a scientific basis for the application of SA as a herbal additive in ruminants.

## 2. Materials and Methods

### 2.1. Experimental Design and Diet Composition

In this trial, a one-way completely randomized experimental design was used to select 24 healthy 3-month-old Dumont crossbred male lambs with similar body weights to be uniformly immunized, dewormed and randomly divided into 3 groups: C1 (the control group), fed a concentrate-to-forage ratio of 50:50; H2 (the high-concentration group), fed a concentrate-to-forage ratio of 70:30; and S3 (the SA group), fed a concentrate-to-forage ratio of 70:30 + 0.1% SA. The experiment was divided with 15 days established as the preliminary period (fulfilling the gastrointestinal tract regarding individual diets) and 60 days as the main period. The experimental diet was designed according to the table of common feed ingredients and the nutritional value of Chinese sheep (NY/T 816-2004), and the diet formula and nutritional ingredients are shown in Table 1.

The source and application method of SA:SA was purchased from Yanchi, Ningxia. After being pulverized and sieved, it was mixed with the concentrate supplement feeding. The content of total alkaloids (main active substances) in SA was 5.75% as determined by acidic dye spectrophotometry, and the recovery rate was 95.95%.

### 2.2. Feeding Management

The feeding trial was carried out in the teaching experimental ranch of Inner Mongolia Agricultural University. The sheep shed was thoroughly disinfected before the trial. During the trial, the test sheep were randomly assigned to 1.5 × 1 × 1 m^3^ iron cages for single-cage feeding according to the principle of non-significant differences in body weight between groups. The humidity and temperature of the sheep shed were kept at 50–60% and 15–20 °C, respectively, with natural ventilation. The test sheep were fed twice a day at 8:00 am and 6:00 pm, with free access to feed and water.

### 2.3. Sample Collection and Determination Indicators

#### 2.3.1. The Rumen pH Value

The rumen fluid was taken from the sheep 0 h before feeding and 2 h, 4 h, 6 h, 8 h and 10 h after feeding using the oral sampling method at the late stage of formal feeding; the fluid was filtered through 4 layers of gauze. The pH of the rumen fluid was determined using a portable pH meter. Specifically, 6 sheep in good condition were selected from each group; 3 sheep from each group were selected for the collection of rumen fluid at 0 h; another 3 were selected for the collection of rumen fluid at 6 h on the first day of the final feeding period; on the second day, rumen fluid was collected at 2 h and 8 h; rumen fluid was collected on the third day at 4 h and 10 h; and on the following three days, the sampling time points of the 3 sheep in each group were interchanged to meet the 6 replicate values in each group.

#### 2.3.2. Liver Function Indicators

Blood samples were collected from the jugular vein of the test sheep on the morning of the 60th day of the feeding period, and the blood was immediately centrifuged at 3000× *g* r/min for 15 min to collect the serum, which was frozen at −20 °C for subsequent determination. ALT, AST, ALP, LDH, ALB and TP were measured using a HITACHI 7020 automatic biochemical analyzer, and the diagnostic kits were purchased from Beijing Lepu Diagnostic Technology Co., Ltd. (Beijing, China).

#### 2.3.3. LPS and LBP in the Serum and Liver and Inflammatory Signaling Factors in the Liver

On the 60th day of the feeding period, 6 sheep with a close-to-average weight were selected from each group for euthanasia, and the tissues from the same part of the left lobe of the lambs’ livers were immediately rinsed with autoclaved double-distilled water and subsequently cut into small pieces with a scalpel to the size of approximately 1 m^3^. Some of the pieces were placed in freezing tubes, immediately put into liquid nitrogen for storage and subsequently put into a refrigerator at −80 °C for later use. The other part was put into the prepared 4% polymethylene glycol for fixation and preservation and was subsequently used for tissue sectioning. Kits for the determination of LPS and LBP contents in the serum and liver as well as IL-1β, IL-6, IL-10, IFN-γ and TNF-α contents in the liver were purchased from Wuhan Genome Biotechnology Co., Ltd. (Wuhan, China).

#### 2.3.4. Liver Antioxidant Indicators

Approximately 0.8 g of an unfrozen liver sample was rinsed with precooled saline to remove blood stains and dried with filter paper. Then, 0.5 g of the liver sample was accurately weighed, and 4.5 mL of precooled saline was added to prepare tissue homogenates using a high-speed homogenizer. The homogenates subsequently were centrifuged at 3000× *g* r/min for 15 min at 4 °C, after which the supernatants were collected for the measurement of antioxidant indices. The T-AOC, CAT, SOD, GSH-Px activity and the MDA content were determined via the colorimetric method using an enzyme marker. The AOC, CAT, SOD, and GSH-Px activities and the MDA content were determined via the colorimetric method. The OD value was read via an enzyme-labeled instrument (model: SYNERGY H11, BioTek, Buelington, Vermont, USA), and the activities of antioxidant enzymes in the liver were expressed as U/mg prot. The kits for the determination of antioxidant indices were obtained from the Nanjing Jiancheng Bioengineering Institute (Nanjing, China). The protein content in the liver tissue was determined via a BCA protein concentration assay kit (enhanced version) provided by the Biyuntian Biotechnology Company (Shanghai, China).

#### 2.3.5. Liver Tissue Sections

Sheep liver tissue sections were prepared according to the methods of Ishikawa et al. [21] and slightly improved upon. The specific steps were as follows: (1) Fixation and washing: The liver tissue was cut into approximately 2.0 × 2.0 cm^2^ small pieces and fixed in 10% formaldehyde for 48 h. Afterward, the tissue was rinsed with tap water to remove residual fixative. (2) Dehydration and transparency: Dehydration was performed using different concentrations of ethanol (60–95%) step by step for 1 h each, followed by two dehydrations of anhydrous ethanol for 45 min, after which the tissue blocks were transferred to xylene for 15 min/until the blocks became transparent. (3) For waxing and embedding, the tissue blocks were put in an equal mixture of melted paraffin wax and xylene to macerate for 1 h and then transferred into paraffin wax for 2 h and poured into the embedding frame. Additionally, the cut side was facing downward, and the embedding wax was poured into the embedding frame. The embedding frame was subsequently placed into the water to cool down as soon as the paraffin wax solidified, and the paraffin wax around the tissue block was kept at a thickness of 2–3 cm. (4) Slicing and baking: Before sectioning, the wax blocks were placed in a refrigerator at −20 °C for 20 min, after which the wax blocks were cut into slices with a thickness of 6 μm using a slicer. The slides were allowed to dry for 12 h and then baked at 65 °C for 30 min. (5) HE staining and sealing were performed. After dewaxing and hydration, the slices were dyed in hematoxylin aqueous solution for 3 min, differentiated with hydrochloric acid and ethanol for 15 s, and washed with tap water for 15 min. Afterward, the slices were dyed in eosin staining solution for 1 min and then dehydrated with absolute ethanol. The sections were cleared with xylene and neutral resin glue was added to the transparent sections, which were placed in a 65 °C oven for 15 min, and after the sections were completely dried, the sections were imaged under a Leica microscope (200×) to observe the pathological changes in the liver tissue.

#### 2.3.6. Liver Transcriptome Sequencing

The extraction and sequencing of RNA from liver tissue were performed. Total RNA was extracted from all samples via the TRIzol method. The purity and concentration of RNA were detected using a Nanodrop nucleic acid analyzer and Qubit. The integrity of the RNA was tested using an Agilent 2100 system. The concentration of each sample was greater than 100 ng/μL, and the RIN was >7.5. A cDNA library was constructed according to the operating instructions of an Illumina TruSeqTM RNA (Illumina, San Diego, California, USA) sample preparation kit. After the samples passed the test, three samples with the best purity and concentration were selected for library construction and sequencing by Shanghai Meiji Biomedical Technology Co., Ltd. (Shanghai, China). The constructed library was sequenced via the Illumina NovaSeq 6000 sequencing platform. After quality control, the original data were compared with the sheep reference genome (Capra hircus, https://www.ncbi.nlm.nih.gov/genome/?term=txid9940[orgn], accessed on 10 April 2021). The expression value of each gene was analyzed using RSEM software (http://deweylab.github.io/RSEM/, accessed on 8 June 2021), and TPM was used as a measure of the gene expression level. The GO and KEGG databases were used to annotate the differentially expressed genes, and the GO and KEGG databases were used to analyze the enrichment of functions and signaling pathways of the differentially expressed genes.

#### 2.3.7. Expression of Differential Genes in Liver Tissue (q-PCR Verification)

Total RNA was extracted from the liver tissue and reverse transcribed into cDNA, which was synthesized according to the instructions of the Prime Script^TM^ RT reagent Kit (Takara Bio Inc., Beijing, China). With respect to the gene sequences of sheep species published in the GenBank of the NCBI website, we used Primer-BLAST in the NCBI database to design gene-specific primers, which were synthesized by Shanghai Sangon Biotechnology Co., Ltd. (Shanghai, China). The primer sequences are shown in Table 2. q-PCR was performed according to the instructions of the TB Premix Dimer Eraser^TM^ kit (Takara Bio Inc., Beijing, China). The sequence of the q-PCR mixture was as follows: 20 µL of TB Green premix (10 µL), 2 µL of upstream and downstream primers, 7 µL of ddH_2_O, and 1 µL of cDNA. The amplification procedure was as follows: step 1, 95 °C for 30 s, a total of 1 cycle; step 2, 95 °C for 5 s, 60 °C for 30 s, a total of 40 cycles; and step 3, 95 °C for 15 s, 60 °C for 30 s, 95 °C for 30 s, a total of 1 cycle. In this study, β-actin was used as the internal reference gene, and the expression level of each target gene was calculated via the 2^−ΔΔCt^ method.

### 2.4. Data Analysis

In this experiment, all the recorded data were entered into an Excel table for preliminary collation. The rumen pH, liver function, immunity, antioxidant indices and relative expression of the genes identified via q-PCR in sheep were analyzed via one-way ANOVA using the GLM in SAS 9.2 and Duncan’s method for multiple comparisons. Significance analysis of the three groups of differentially expressed genes was performed using Tukey’s one-way ANOVA with SPSS Statistics 17 software. *p* < 0.05 was considered to indicate statistical significance, *p* < 0.01 was considered to indicate high significance, and 0.05 < *p* < 0.1 was considered to indicate a significant trend. The differential genes were analyzed for GO and KEGG enrichment using GOATOOLS software (https://github.com/tanghaibao/GOatools, accessed on 10 June 2021) and P scripts (https://scipy.org/install/, accessed on 10 June 2021), respectively, and tested for precision via Fisher’s method. The corrected *p* value (*p*-adjust) was calculated using the Benjamini–Hochberg method (BH). GO and KEGG functional enrichment data were considered significant when *p*-adjust < 0.05 indicated a significant enrichment, and 0.05 < *p*-adjust < 0.1 indicated a trend toward significant enrichment.

## 3. Results

### 3.1. Effect of Adding SA to High-Concentrate Diets on the Rumen pH of Sheep

As shown in Figure 1, at 0 h (before feeding) and after 10 h of feeding, there was no significant difference in the rumen pH among the three groups of sheep (*p* > 0.05). The rumen pH value of the S3 group was significantly higher than that of the H2 group at 2–4 h of feeding (*p* < 0.05). The pH values of the C1 and S3 groups were significant or extremely significant compared to those of the H2 group at 6–8 h of feeding (*p* < 0.05 or *p* < 0.01). The rumen pH of the sheep in the C1 group tended to decrease and then increase with increasing feeding time, while the pH of the H2 group decreased markedly with increasing feeding time, reaching the lowest value of 5.44 at 2 h of feeding and starting to increase after 8 h. The pH of the S3 group changed gradually.

### 3.2. Effect of Adding SA to High-Concentrate Diets on Liver Function Indices in Sheep

As shown in Figure 2a–f, the serum ALT activity was significantly higher (*p* < 0.01) in the H2 group than in the C1 and S3 groups and was significantly higher (*p* < 0.05) in the S3 group than in the C1 group. AST and LDH activities were extremely significantly higher (*p* < 0.01) in the H2 group than in the C1 and S3 groups. The TP content was significantly higher (*p* < 0.05) in the C1 group than in the H2 and S3 groups. ALP and ALB concentrations did not significantly differ among the three groups (*p* > 0.05).

### 3.3. Effect of Adding SA to High-Concentrate Diets on LPS and LBP in the Serum and Liver and Inflammatory Signal Factors in the Liver of Sheep

As shown in Figure 3a,b, the LPS and LBP concentrations in the serum of sheep were significantly higher in the H2 group than in the C1 and S3 groups (*p* < 0.05), the LPS concentration in the liver was significantly higher in the H2 group than in the S3 group (*p* < 0.05), and the LBP concentration in the liver was significantly higher in the H2 group than in the C1 and S3 groups (*p* < 0.01). As shown in Figure 4a–e, the IL-6 content in the liver of the sheep in the C1 and H2 groups was significantly higher than that in the S3 group (*p* < 0.01), and the IL-10 content in the liver of the sheep in the C1 and S3 groups was significantly higher than that in the H2 group (*p* < 0.01). The TNF-α content in the liver tissue of the sheep in the S3 group showed a decreasing trend compared with the H2 group (*p* = 0.086), and the IL-1β and INF-γ contents in the liver tissue of the sheep were not significantly different among the three groups (*p* > 0.05).

### 3.4. Effect of Adding SA to High-Concentrate Diets on the Liver Antioxidation of Sheep

As shown in Figure 5a–e, the CAT activity in the liver of the sheep in the C1 group was significantly higher than that in the H2 group (*p* < 0.05), and there was no significant difference between the H2 group and the S3 group (*p* > 0.05). The SOD activity in the liver of the sheep in the H2 group was significantly lower than that in the C1 and S3 groups (*p* < 0.05). The GSH-P_X_ and T-AOC activities in the sheep liver in the S3 group were significantly or extremely significantly higher than those in the C1 and H2 groups (*p* < 0.05 or *p* < 0.01). There was no significant difference in the MDA content in the sheep liver among the three groups (*p* > 0.05).

### 3.5. Effect of SA Addition to High-Concentrate Diets on Sheep Liver Tissue Sections

As shown in Figure 6, in the C1 group, the structure of the liver lobules was clear, and the hepatocyte cords were neatly arranged, radiating around the central vein. The hepatocytes were polygonal, uniform in size, and occasionally binucleated, and no obvious vacuoles were observed in the cells. In the H2 group, the structure of the liver lobules was damaged, the structure of the hepatocyte cords was disorganized, there was an infiltration of inflammatory cells, mainly lymphocytes and vacuolated hepatocytes, and some of the hepatocytes showed eosinophilic changes with deep nuclear staining. In the S3 group, the structure of the hepatocyte cords was disorganized, vacuolated hepatocytes were reduced, binucleated cells were observed, and a few of them exhibited eosinophilic changes with deep nuclear staining; these changes in the S3 group were less severe than those in the H2 group. Regenerative hepatocytes were observed.

### 3.6. Effect of Adding SA to High-Concentrate Diets on the Sheep Liver Transcriptome

#### 3.6.1. Quality Control of Liver Transcriptome Data

As shown in Table 3, the completion of the transcriptome analysis of nine samples revealed that the sample error rate was approximately 0.025%, all below 0.1%, and the percentage of bases with a sequencing quality of 99% and 99.9% or more of the total bases was above 90.0%. The clean reads of each sample were sequenced against the designated reference genome separately, and the average comparison rate was 85.51 ± 0.76%. A total of 26,478 genes were detected in the liver transcriptome, 10,009 of which were expressed at ≥1 level in the sheep liver.

#### 3.6.2. Sample Relationship Analysis

We constructed a heatmap via Pearson correlation analysis (Figure 7) and found that the red regions accounted for most of the samples, with fewer light blue regions; moreover, the correlation coefficients (R^2^) between the samples were greater than 0.83, indicating that there was a strong correlation between the samples (R^2^ = 0.8–1.0), that there was very good stability of the sequencing among the biological replicates, and that the sequencing data could be used for subsequent analyses.

#### 3.6.3. Differential Gene Analysis of the Three Groups

A total of 519 differentially expressed genes were obtained in the three groups through multigroup expression difference analysis, and we selected 10 differential genes related to immunity and oxidative stress based on the above liver immunity and antioxidant indices for bar graph plotting (Figure 8). Hepatic *SOD2* expression was significantly or extremely significantly higher in the C1 and S3 groups than in the H2 group (*p* < 0.05 or *p* < 0.01). The expression levels of *IRF5*, *TNFSF18*, *PROCR* and *PLK1* were significantly higher in the H2 group than in the C1 and S3 groups (*p* < 0.05); the expression levels of *SQOR*, *CD93* and *FOXO4* were significantly higher in the C1 and H2 groups than in the S3 group (*p* < 0.05); and *TGFB2* expression was significantly higher in the C1 group than in the H2 and S3 groups (*p* < 0.05). *IL27RA* expression was significantly higher in the H2 subgroup than in the S3 subgroup (*p* < 0.05).

#### 3.6.4. Effect of Liver Gene Function in Sheep

We further analyzed and summarized the GO and KEGG (top 20) functional classifications of the differentially expressed genes in the three groups (Figure 9a,b). GO functional annotation analysis revealed that the differentially expressed genes associated with biological processes (BPs) were involved in eight secondary classification processes, the cellular process, biological regulation and the metabolic process. The differentially expressed genes in the cellular component (CC) category were involved in eight secondary categorization components such as cell parts, organelles and organelle parts. In the molecular function (MF) category, genes were involved in four main components, namely, binding, catalytic activity, and transcription regulator activity. KEGG functional annotation analysis showed that the differentially expressed genes involved in metabolism mainly participated in glycine biosynthesis and metabolism, whereas those involved in genetic information processing mainly participated in replication and repair. The differentially expressed genes involved in environmental information processing mainly participated in signaling molecules and interactions and signal transduction, and the genes involved in cellular processes mainly participated in cell growth and death, cellular community—eukaryotes, transport and catabolism. The differentially expressed genes in organismal systems were involved mainly in the immune system, the endocrine system, the digestive system and the nervous system. In human diseases, these genes were involved in nine secondary classification functions such as infectious disease (viral), cancer (overview), and infectious disease (bacterial).

#### 3.6.5. GO and KEGG Enrichment Analyses of Sheep Liver Data

We filtered the top 20 GO terms associated with enrichment according to the *p*-adjust < 0.1 condition (Figure 10a) and found that, in the CC category, differentially expressed genes were significantly enriched in the collagen trimer, the fibrillar collagen trimer, the extracellular matrix, the external encapsulating structure, and the collagen-containing extracellular matrix (*p*-adjust < 0.01), and there was a trend toward significant enrichment in the extracellular region (*p*-adjust = 0.069). The differentially expressed genes included mainly *CRABP1*, *IL13*, *CCL17*, *GDF2*, *TGFB2,* and others. In the BP category, the differentially expressed genes were significantly enriched for collagen fibril organization, external encapsulating structure organization, extracellular matrix organization, extracellular structure organization, animal organ morphogenesis, anatomical structure morphogenesis, positive regulation of the cellular process, the collagen metabolic process, the cellular response to chemical stimulus and the response to chemical (*p*-adjust < 0.05 or *p*-adjust < 0.01), which, in the developmental process and embryonic morphogenesis, had a tendency to be significantly enriched (*p*-adjust = 0.084 or *p*-adjust = 0.087). The differentially expressed genes included *SOD2*, *PLK1*, *FOXO4*, *TGFB2*, *TNFSF18*, *IRF5*, *IL27RA*, and others. In the MF category, the differentially expressed genes were significantly enriched in the platelet-derived growth factor binding process (*p*-adjust < 0.01), and there was a trend toward significant enrichment in the extracellular matrix structural constituent conferring tensile strength (*p*-adjust = 0.058). The differentially expressed genes included *COL3A1*, *COL1A1*, *PDGFB*, and *COL1A2*. Among them, the fibrillar collagen trimer, the extracellular matrix structural constituent conferring tensile strength, the platelet-derived growth factor binding process, collagen fibril organization and the collagen metabolic process were the most enriched.

We found that the differentially expressed genes were significantly enriched in the ECM–receptor interaction, focal adhesion and pyrimidine metabolism via KEGG enrichment analysis (Figure 10b). Among them, the ECM–receptor interaction and focal adhesion pathways were closely related to immune and antioxidant functions.

#### 3.6.6. Analysis of Differentially Expressed Genes in Immune- and Antioxidant-Related Pathways in Sheep Liver

As shown in Table 4, the expression levels of *COL1A1*, *COL1A2*, *COL6A2*, *COL6A3* and *PAK6* in the liver were significantly or extremely significantly lower in the C1 and S3 groups than in the H2 group (*p* < 0.05 or *p* < 0.01). *COL6A5* and *LAMC1* expression levels in the liver were significantly lower in the S3 group than in the H2 group (*p* < 0.05). *ARHGAP35* and *VCL* expression levels in the liver were significantly lower in the S3 group than in the C1 and H2 groups (*p* < 0.05). *ITGA5* expression in the liver of the C1 group was significantly lower than that in the H2 and S3 groups (*p* < 0.05), and *PDGFB* expression in the liver of the C1 and S3 groups was significantly or extremely significantly lower than that in the H2 group (*p* < 0.05 or *p* < 0.01), while that in the S3 group was significantly lower than that in the C1 group (*p* < 0.05). The differences in expression between the other groups were not significant (*p* > 0.05).

### 3.7. q-PCR Verification of Differential Gene Expression

As shown in Table 5, the relative expression levels of the *IL-6*, *TGFB2* and *ITGA5* mRNAs in the sheep liver were significantly higher in the H2 and S3 groups than in the C1 group (*p* < 0.01). Moreover, there were no significant differences in the relative *INF-γ*, *GPX1* or *IL-1β* mRNA expression levels among the three groups (*p* > 0.05). The relative *TNF-α* mRNA expression level in the sheep liver in the H2 group was significantly higher than that in the S3 group (*p* < 0.05), but there was no significant difference between the H2 group and the C1 group (*p* > 0.05). The relative *CD93* and *PAK6* mRNA expression levels in the sheep liver in the H2 group were significantly or extremely significantly higher than those in the C1 and S3 groups (*p* < 0.05 or *p* < 0.01). The *SOD2* mRNA relative expression level in the sheep liver in the H2 group was significantly or extremely significantly lower than that in the C1 and S3 groups (*p* < 0.05 or *p* < 0.01), and the *VCL* mRNA relative expression level in the C1 and H2 groups was extremely significantly higher than that in the S3 group (*p* < 0.01).

## 4. Discussion

The ruminal pH value is an important indicator of normal rumen fermentation function, and the current diagnosis and definition of SARA depend on the indicator of ruminal pH. The normal pH range of the rumen in ruminants is 6.0–7.5, and a decreased rumen pH (5.2–5.6) that persists for 3 h is often recognized as the cause of SARA [22,23]. The pH is affected by the feeding method, the concentrate-to-forage ratio and the dry matter content, and a substantial decrease in the rumen pH is the most common feature in the rumen of ruminants fed high-concentrate diets. Many studies have found that a decrease in pH to approximately 6.0, after animals regularly ingest a high-grain diet, results in a slight decrease in the fiber decomposition rate; however, the number of fibrolytic bacteria is usually unaffected, and a decrease to 5.5 or 5.0 results in a decrease in both the fiber decomposition rate and the number of fibrolytic bacteria, which may severely inhibit fiber digestion altogether, thus altering the rumen fermentation function [24,25]. In this study, the pH of the H2 group decreased sharply with feeding time and reached a minimum value of 5.44 after 2 h of feeding. The pH was at 5.61 after 6 h of feeding and began to increase after 8 h, suggesting that there was a possibility of SARA occurring in the high-concentrate group. The pH of the S3 group was higher than that of the H2 group, and the changes were relatively smooth; moreover, the rumen pH values were all approximately 6.0, indicating that the addition of SA to high-grain diets was beneficial for improving the rumen pH and might play an important role in the microbial utilization of cellulose to change the rumen fermentation function. These findings were also verified by our previous results showing that SA increased the relative abundance of *Bacteroides* in the rumen (fiber- and starch-decomposing bacteria) and decreased the relative abundance of *Firmicutes* (acid-resistant bacteria) [17].

Under high-concentrate diets conditions, the rumen pH decreases and gram-negative bacteria proliferate, accumulating large amounts of LPS as they grow. When the rumen becomes too acidic, some of the bacteria cannot continue to survive, and the gram-negative bacteria shed and disintegrate, releasing large amounts of free LPS [26,27]. A portion of free LPS enters the bloodstream from the gastrointestinal tract and then enters the liver through the portal vein, whereas uncleared LPS in the liver induces structural and functional changes in the liver [28] and stimulates the secretion of proinflammatory cytokines by hepatocytes and lymphocytes, which in turn activates the inflammatory response [29]. The other part of the LPS complex binds to LBP (LBP is a glycoprotein synthesized in hepatocytes and secreted into the serum) on the cell membrane to form a complex that acts on cell membrane receptors to promote the release of inflammatory mediators such as TNF-α, NO and IL-6 through MAPK, NF-κB and other signaling pathways, thereby triggering inflammatory responses in the liver and causing indirect damage to the liver [30,31]. In the present study, the LPS and LBP contents in the blood and liver were markedly higher in the H2 group than in the C1 and S3 groups, indicating that LPS-induced production by high-concentrate diets causes a systemic inflammatory response. The lowest level of LPS was found in the S3 group, indicating that SA helps to alleviate direct or indirect damage to the liver caused by LPS in sheep. The total alkaloids of SA and the three monomeric alkaloids of SA can alleviate pathological damage in mice with LPS-induced lung injury to varying degrees, and the anti-LPS mechanism may be related to regulating the recognition receptor of LPS and affecting the expression of downstream inflammatory factors [32]; moreover, the specific mechanism involved needs further exploration.

Liver function is an important indicator of the anabolic function of the liver. Animals have low serum concentrations of ALT, AST, and LDH under normal conditions, but when liver cells or tissues are damaged, ALT, AST, and LDH are released into the bloodstream and are elevated [33]. Matrine injection inhibited ALT and AST in mouse serum during liver injury and increased GSH, CAT and SOD activities, which suggested that matrine injections could regulate serum transaminase levels, improve liver function, increase antioxidant enzymes and improve oxidative stress [34]. Although the increase in the serum LDH concentration is not specific to any single tissue or organ, its activity can reflect the stress sensitivity of animals. SA improved the survival rate of SI/R-inhibited H9c2 cardiomyocytes and inhibited LDH levels in cardiomyocytes, thus reducing the occurrence of myocarditis [35] and indicating that SA can also reduce the stress sensitivity of animals. In this study, the ALT, AST and LDH activities in the serum of the sheep in the C1 and S3 groups were lower than those in the H2 group, which was consistent with the above findings. Overall, these findings indicate that the addition of SA to high-concentrate diets can regulate liver function indices, reduce the stress sensitivity of animals, and subsequently exert a certain protective effect on the liver. If 60–80% of animal liver parenchymal function is lost, the synthesis of ALB will be affected, and the protein content in plasma will be reduced. Therefore, the health status of liver function can be judged by clinically measuring and detecting the protein content in plasma [36]. The serum ALB content in this study was not significantly different among the three groups, but it was lower in the S2 subgroup. Consistent with the findings of Li et al. [37], who showed no significant change in plasma protein after LPS injection into the extravaginal arteries of dairy cows, these findings prove that the degree of liver damage at 60 days of high-concentrate feeding to the sheep does not lead to a blockage of protein synthesis; however, this change may also be related to the feeding conditions, individual differences, and time of blood collection from the sheep.

It is well known that inflammatory cytokines are important indicators of the immune response in animals; they mediate T, B and other cell activation, proliferation and differentiation and play important roles in the inflammatory response. SA alkaloids inhibited the LPS-induced secretion of proinflammatory cytokines, such as IL-6 and TNF-α, in rats in a dose-dependent manner [38,39], suggesting that SA alkaloids have an inhibitory effect on the inflammatory responses caused by LPS stimulation. Liang et al. [40] also reported that, compared with those in the LPS model group, the pathological injury to lung and kidney tissues was reduced to different degrees and that the serum NO, TNF-α, and BUN levels were decreased. Additionally, the protective effect of sophoridine on LPS-induced mice may be related to the inhibition of inflammatory factor release, and the protective effect of sophoridine on endotoxemic mice may be related to the inhibition of inflammatory factor release. In the present study, the IL-6 and TNF-α contents in the liver of sheep in the C1 group were higher than those in the S3 group, while the change in the IL-10 content in the liver of sheep was the opposite. Overall, these findings indicate that 0.1% SA could effectively inhibit the secretion of proinflammatory cytokines and increase the concentration of anti-inflammatory cytokines to improve the immune function of the lamb liver and to alleviate the inflammatory response. This finding is similar to our previous findings on the serum inflammatory factor profile.

Previous reports have shown that LPS exposure can result in oxidative stress by increasing ROS formation [41]. A large number of studies have shown that, in vivo, the mitochondrial MDA content of hepatocytes is elevated, and the SOD activity is decreased in rats injected with LPS, which increases the generation of endogenous oxygen radicals, affects the respiratory function of mitochondria, and results in oxidative damage to the mitochondria of hepatocytes [42]. Yu et al. [43] used a combination of BCG and LPS to establish a mouse immune liver injury model and then gavaged BALB/c mice with high-dose matrine (80 mg/kg), which led to a decrease in the MDA and NO levels and an increase in the T-AOC as well as GSH-Px activity; additionally, the combination of these agents significantly improved the activity of antioxidant enzymes and the elimination of free radicals. These findings demonstrated that high-dose matrine has a strong antioxidant effect and can antagonize free radicals in the liver. In the present study, the activity of SOD in the sheep liver was the lowest in the H2 group, and the GSH-P_X_ and T-AOC activities in the sheep liver were higher in the S3 group than in the C1 and H2 groups. These findings indicate that feeding sheep a high-concentrate diet results in the production of a large amount of free LPS in the liver, which induces oxidation in the sheep’s body and exacerbates stress hazards in lambs, while SA can alleviate this stress and has a stable and efficient antioxidant effect on lambs. 

In this study, inflammatory cell infiltration was found in the liver tissue sections of sheep in the high-concentrate group, mainly involving lymphocytes and eosinophils; the swelling and degeneration of liver cells; and the dysfunction of the liver cord. However, in the SA group, inflammatory cells occasionally infiltrated the liver, and the number of swollen hepatocytes was reduced; only a few hepatocytes exhibited eosinophilia with deep nuclear staining, which indicated that 0.1% SA could effectively alleviate inflammatory damage in the liver. Surprisingly, after BALB/c mice were injected with 4, 8 or 16 mg/kg of aloperine for 4 weeks, Qiu et al. [44] found that the cytoplasm of hepatocytes in the high-dose group was vacuolated, the epithelial cells of renal tubules were swollen, and the expression and activity of CYP450 changed; however, all the cells recovered to some extent after 1 week without the use of aloperine, and all the cells recovered completely after 4 weeks. These findings showed that aloperine has a reversible toxic effect on mice. We found that, in the feeding process, adding 0.5% of SA to the diet of lambs will result in the manifestation of urinary frequency and other phenomena. Therefore, this suggests that the selection of the amount of SA additives in the diet of sheep is of vital importance. Moreover, in a previous study, we confirmed that 0.1% SA is the optimal dosage.

Our functional classification analysis of liver differentially expressed genes revealed that the 519 differentially expressed genes were involved mainly in cellular processes, biological regulation, metabolic processes, cell parts, organelles, organelle parts, binding catalytic activity, and transcription regulator activity, etc., which were involved mainly in glycan biosynthesis and metabolism, replication and repair, signaling molecules and interactions, signal transduction, cell growth and death, cellular community—eukaryotes, transport and catabolism, the immune system, the endocrine system, the digestive system, and other pathways. In addition, we analyzed the GO functional enrichment of liver differentially expressed genes and found that 519 differentially expressed genes were significantly enriched in the collagen trimer, the fibrillar collagen trimer, the extracellular matrix, the external encapsulating structure, collagen fibril organization, external encapsulating structure organization, extracellular matrix organization, extracellular structure organization, animal organ morphogenesis, the platelet-derived growth factor binding process, and the extracellular matrix structural constituent conferring tensile strength and other functions. The differentially expressed genes included mainly *CRABP1*, *IL13*, *CCL17*, *GDF2*, *TGFB2*, *SOD2*, *PLK1*, *FOXO4*, *TNFSF18*, *IRF5*, *IL27RA*, *COL3A1*, *COL1A1*, *PDGFB*, and *COL1A2*. Additionally, 10 genes related to immunity and antioxidants, that were previously selected, were significantly enriched in GO terms in both the CC and BF categories, suggesting that the biology of immunity and antioxidant-related issues in the liver is the focus of our research.

KEGG enrichment analysis demonstrated that the ECM–receptor interaction and focal adhesion pathways were closely related to immune and antioxidant functions, and the differentially expressed genes enriched in both pathways were essentially the same. The ECM consists of a complex mixture of structural and functional macromolecules and plays important roles in connecting different parts of the immune and antioxidant response [45]. The ITGA5 and ITGA6 proteins belong to the integrin family and act as transmembrane signaling molecules involved in cell adhesion and signaling between the ECM and cells [46]. One study revealed that the knockdown of the Twist2 gene suppressed the expression of ITGA6 and CD44, which are involved in the migration and invasion of cancer cells and are associated with the ECM–receptor interaction pathway. Twist2 may promote the migration and invasion of renal cancer cells by regulating the expression of ITGA6 and CD44 [47]. CD93 belongs to the C-type lectin domain (CTLD) group 14 family of transmembrane glycoproteins together with thrombomodulin and CD248. This protein family has a similar ectodomain architecture to that of CD44 and is involved in several cell processes including angiogenesis, inflammation, tumor development and cell adhesion [48,49]. Previous reports have shown that recombinant sCD93 can induce cell adhesion and increase phagocytic activities. This results in an enhanced response to stimulation by TLRs via LPS, increasing the production of proinflammatory cytokines such as IL-6 and IL-1β [50]. In this study, *ITGA5* expression in the C1 subgroup was lower than that in the H2 and S3 subgroups, but this difference was not significant in the H2 and S3 groups, which was consistent with our q-PCR verification results. Overall, these findings indicate that SA may not regulate the ECM–receptor interaction pathway through the expression of *ITGA5* mRNA, but these findings should also be considered at the protein level. In addition, we found that *CD93* expression was the highest in the H2 group and the lowest in the S3 group. These findings indicated that the increase in LPS promoted the expression of *CD93* mRNA under high-grain conditions and further induced cell adhesion. The activated TLRs pathway promoted the production of proinflammatory factors, while SA may inhibit the LPS-induced liver inflammatory reaction by blocking the TLRs/NF-κB pathway, which is similar to our previous finding on the anti-inflammatory pathway in the gastrointestinal tract [51]. Although the *CD93* gene was not significantly enriched in these two pathways, it was involved in the process of liver inflammation, which may be caused by the *p*-adjust, as indicated by our screening. FOXO4-D-Retro-Inverso (FOXO4-DRI), a synthetic peptide that perturbs the interaction of FOXO4 with p53, has been found to selectively induce the apoptosis of senescent cells [52]. In senescent cells, the FOXO4 protein was upregulated after treatment with TGF-β, while FOXO4-DRI had a killing effect on TGF-β-induced myofibroblasts. FOXO4-DRI reversed the changes in protein expression, which could be attributed to the clearance of FOXO4-enriched cells. Furthermore, FOXO4-DRI inhibited the upregulation of the α-SMA and COL1A1 genes and proteins. COL1A1 is a component of the ECM that is upregulated during TGF-β-induced myofibroblast differentiation [53]. Therefore, FOXO4-DRI can not only eliminate aging cells and reduce inflammatory stimulation to neighboring cells to ameliorate pulmonary fibrosis (PF) induced by bleomycin (BLM) but can also downregulate the expression of the major ECM genes and proteins, reduce the formation of ECM, and ultimately inhibit ECM–receptor interactions, thus alleviating PF induced by BLM [54]. In this study, for a variety of collagen families, the expression levels of *COL1A1* and *COL1A2* in the S3 group were significantly lower than those in the H2 group. Furthermore, the *FOXO4* expression was lower in the S3 group than in the H2 group, which proved that the addition of SA to high-concentrate diets could reduce the inflammatory stimulation of hepatocytes and alleviate inflammatory injury in the liver by downregulating the expression of major genes and proteins in the ECM and inhibiting the ECM–receptor interaction pathway.

The PAK family of serine/threonine protein kinases, based on their structure and sequence homology (group I and group II), are the chief mediators of CDC42 and RAC, which are important mediators of many cellular functions [55]. Group-II PAKs, including PAK6, have been particularly well studied in neoplastic processes (regarding stomach and other GI tumors), where they have been shown to be overexpressed in many tumors [56]. Previous reports have shown that PAK6 not only functions downstream of FAs signaling but is also proximally recruited to activated EGFR signaling complexes, which probably function upstream of FAs [57]. FAK is a highly conserved nonreceptor tyrosine protein kinase and a key enzyme in the focal adhesion pathway that plays a role in regulating the adhesion of cells to the ECM [58]. FAK can also regulate immune and antioxidant processes by participating in multiple downstream signaling pathways [59,60]. After integrin attaches to the ECM, FAK is activated by growth factor stimulation and promotes cell proliferation through MAPK or PI3-K activation [61]. Additionally, EGF stimulation can enhance the formation of complexes between EGFR, SRC3&4 (EGFR-FAK bridging protein), FAK and PAK, further activating the FAs signaling pathway [57]. During the PDGF-induced proliferative response of hepatic stellate cells, blocking FAK and PAK activities significantly inhibits tumor cell proliferation and PI3-K activity. This signaling occurs mainly through the FAK/PI3K/Akt pathway [62]. In this study, the expression levels of *PAK6*, *ARHGAP35* and *PDGFB* were lower in the S3 group than in the H2 group. Our q-PCR results coincided with the transcriptome results, suggesting that the alleviation of liver injury in sheep via the addition of SA to high-concentrate diets may exert anti-inflammatory and antioxidant effects by downregulating the expression of major genes and proteins in the FAs signaling pathway, further inhibiting the activity of MAPK or PI3-K and indirectly modulating the production of inflammatory cytokines. We will further verify our study at the protein level in the future.

## 5. Conclusions

The addition of SA to high-concentrate diets reduced the ALT, AST, LDH, LPS and LBP activities and concentrations in the serum, decreased the IL-6, TNF-α, LPS and LBP levels in the liver, and increased the IL-10, SOD, GPX-P_X_ and T-AOC levels and activities in the liver of lambs. SA reduced the number of enlarged hepatocytes and inflammatory cell infiltration and restored the regenerative capacity of hepatocytes. The differentially expressed genes affected by SA were involved in functions and pathways such as cellular processes, cell parts, transcriptional regulator activity, signaling molecules and interactions, signal transduction, cell growth and death, and the immune system and were significantly enriched in functions such as collagen fiber organization, extracellular matrix, and platelet-derived growth factor binding. The addition of SA to high-grain diets could modulate the mechanism of liver injury in sheep by regulating the expression of the relevant genes involved in ECM–receptor interactions and focal adhesion pathways.

## Figures and Tables

**Figure 1 animals-14-00182-f001:**
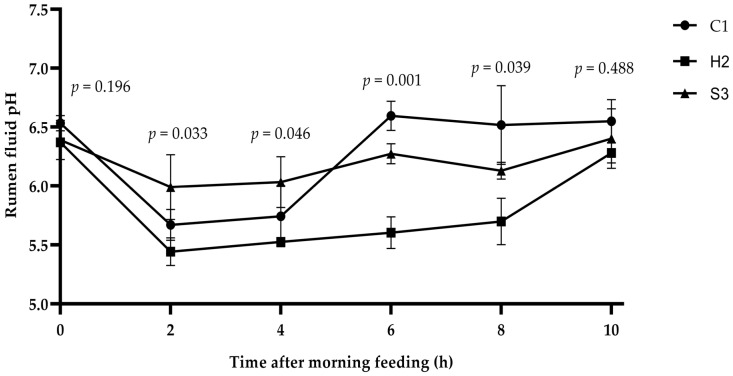
Dynamics of the rumen pH in lambs. Data are means ± SD for *n* = six lambs per group. SD: standard deviation.

**Figure 2 animals-14-00182-f002:**
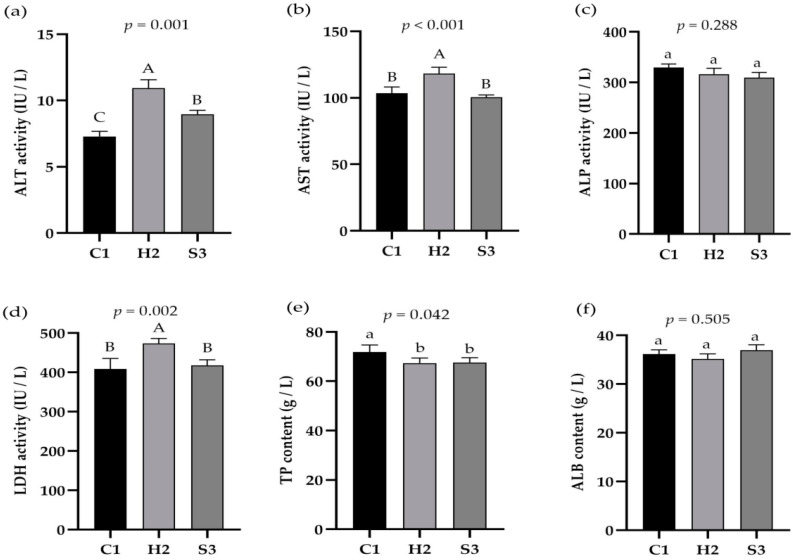
Effect of SA on liver function indices in sheep. (**a**) ALT activity, (**b**) AST activity, (**c**) ALP activity, (**d**) LDH activity, (**e**) TP content, (**f**) ALB content. The data are presented as the means ± SEMs for *n* = six lambs per group. Data marked without the same lowercase letters indicate significant differences at *p* < 0.05, and data without the same capital letters indicate extremely significant differences at *p* < 0.05.

**Figure 3 animals-14-00182-f003:**
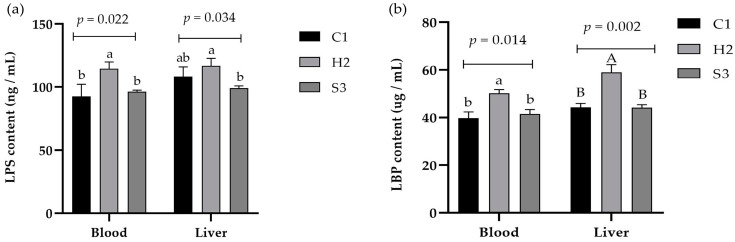
Effect of SA on LPS and LBP in the serum and liver of the sheep. (**a**) LPS content, (**b**) LBP content. Data marked without the same lowercase letters indicate significant differences at *p* < 0.05, and data without the same capital letters indicate extremely significant differences at *p* < 0.05.

**Figure 4 animals-14-00182-f004:**
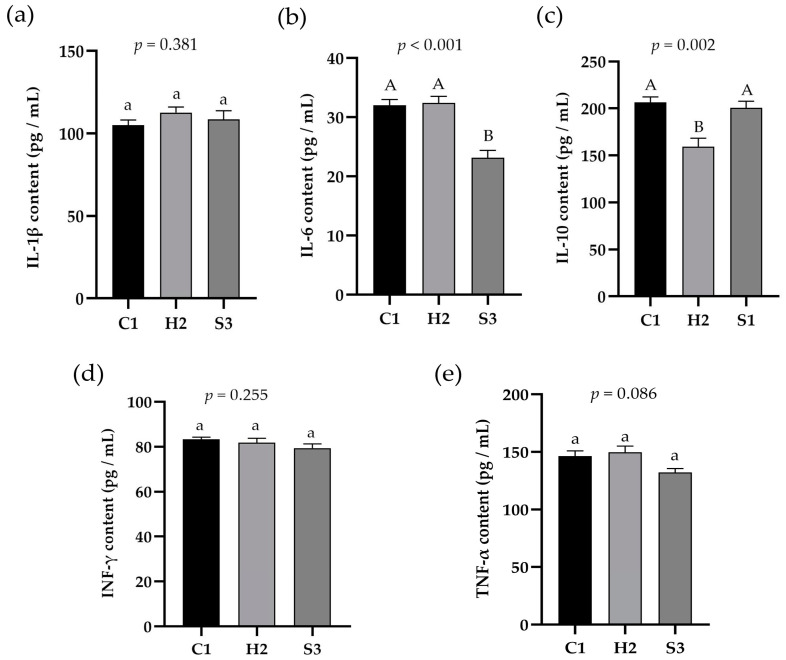
Effect of SA on inflammatory signal factors in the liver of the sheep. (**a**) IL-1β content, (**b**) IL-6 content, (**c**) IL-10 content, (d) INF-γ content, (**e**) TNF-α content.Data marked without the same lowercase letters indicate significant differences at *p* < 0.05, and data without the same capital letters indicate extremely significant differences at *p* < 0.05.

**Figure 5 animals-14-00182-f005:**
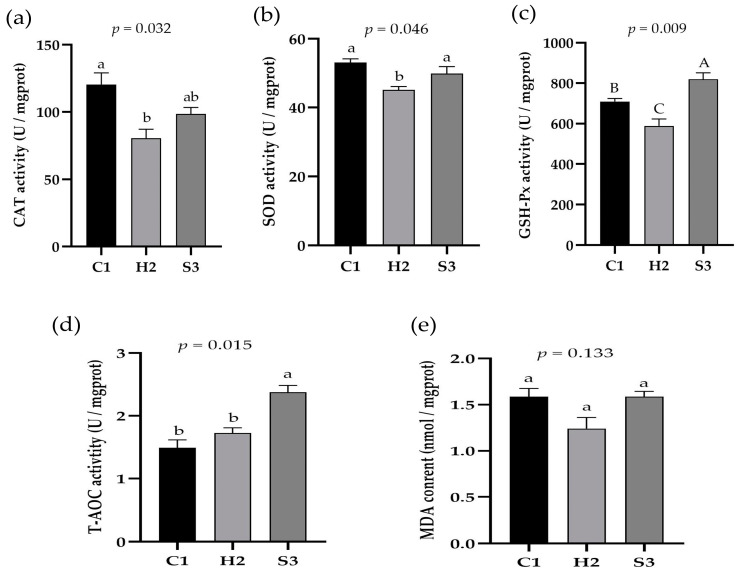
Effect of SA on the liver antioxidation in sheep. (**a**) CAT activity, (**b**) SOD activity, (**c**) GSH-Px activity, (**d**) T-AOC activity, (**e**) MDA content. Data marked without the same lowercase letters indicate significant differences at *p* < 0.05, and data without the same capital letters indicate extremely significant differences at *p* < 0.05.

**Figure 6 animals-14-00182-f006:**
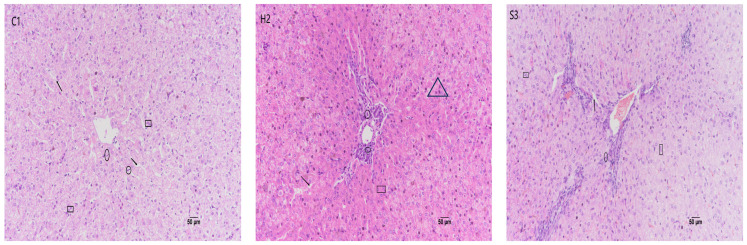
Effect of SA on liver histopathologic changes in sheep (HE × 200). The arrows in the figure indicate hepatocyte cords, the squares indicate binucleated hepatocytes, the ovals indicate eosinophils, and the triangles indicate inflammatory cells.

**Figure 7 animals-14-00182-f007:**
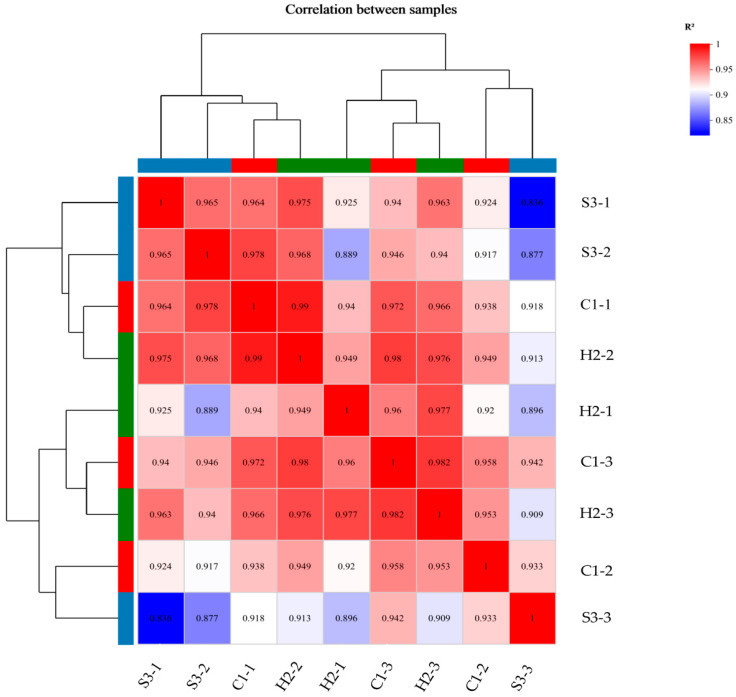
Heatmap of correlation analysis between samples. The right and lower sides are sample names, the left and upper sides are sample clustering, and squares with different colors represent the correlation between the two samples.

**Figure 8 animals-14-00182-f008:**
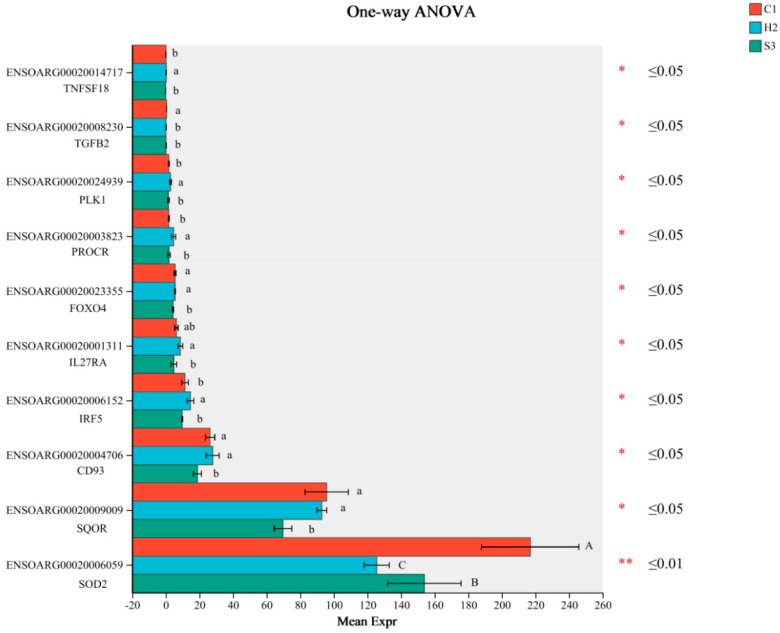
Differentially expressed genes related to immunity and antioxidation in the sheep liver. The Y-axis indicates the gene ID and gene name, the X-axis indicates the average relative expression of genes in different groups, different colored bars indicate different groups, and the right is the *p*-value. The data are presented as the means ± SD of *n* = three lambs per group. Data marked without the same lowercase letters indicate significant differences at *p* < 0.05, and data without the same capital letters indicate extremely significant differences at *p* < 0.05.

**Figure 9 animals-14-00182-f009:**
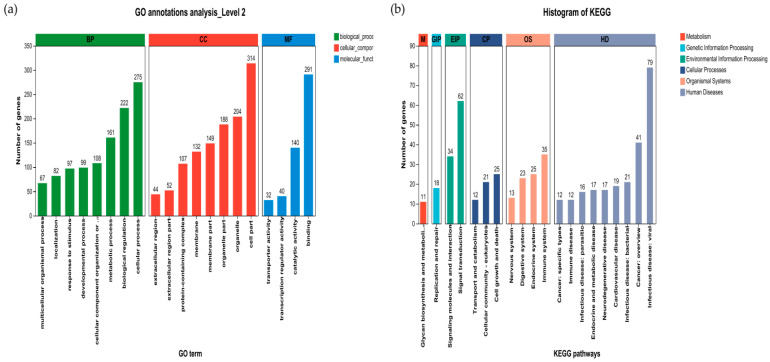
GO and KEGG functional annotation analyses of the differentially expressed genes. The horizontal coordinate indicates the secondary classification term for GO, the vertical coordinate indicates the number of genes in that secondary classification of the ratio, and the three colors indicate the three major classifications (**a**). The horizontal coordinate represents the name of the KEGG metabolic pathway, the vertical coordinate represents the number of genes annotated to the pathway, and the seven categories of KEGG metabolic pathways are shown in different colors (**b**).

**Figure 10 animals-14-00182-f010:**
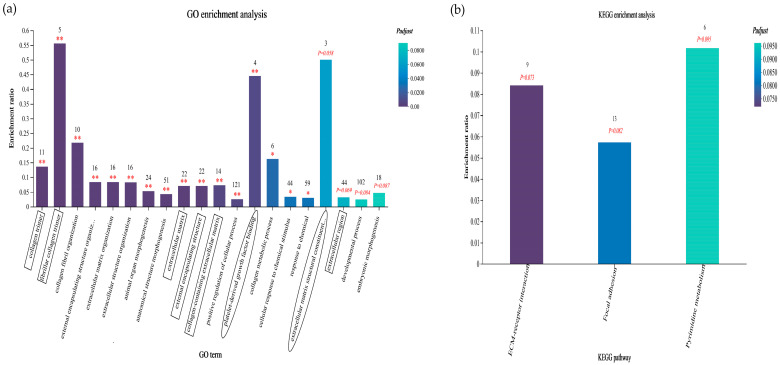
GO and KEGG enrichment analyses of the differentially expressed genes. The number on the top of the bar indicates the number of genes enriched in the GO term, the horizontal coordinate indicates the GO term, the rectangle on the GO term indicates CC, the circle indicates MF, and the absence of markers indicates BP. The vertical coordinate indicates the enrichment factor: the ratio of the number of genes enriched in the GO term to the number of annotated genes; the larger the enrichment factor, the greater the degree of enrichment (**a**). The number on the top of the bar indicates the number of genes enriched in the KEGG pathway, the abscissa represents the KEGG pathway, and the ordinate represents the enrichment factor (**b**). The color gradient indicates the significance of enrichment, where *p*-adjust < 0.01 is marked as **, *p*-adjust < 0.05 is marked as *, and 0.05 < *p*-adjust < 0.1 is marked as the *p*-adjust value.

**Table 1 animals-14-00182-t001:** Composition and nutrient levels of basal diets (DM basis).

Ingredients	Composition, %		Items	Nutrient Levels	
	C1	H2		C1	H2
Mixed hay	50.00	30.00	Metabolizable energy ^(2)^, MJ/kg	9.53	10.20
Corn	36.10	52.00	Crude protein, %	14.17	14.45
Soybean meal	12.00	15.20	Ether extract, %	2.84	3.72
Limestone	0.60	1.00	Neutral detergent fiber, %	33.18	25.03
CaHPO_4_	0.10	0.10	Acid detergent fiber, %	22.36	15.37
NaCl	0.50	0.50	Nonstructural carbohydrate, %	43.6	49.9
Premix ^(1)^	0.20	0.20	Ca, %	0.77	0.71
NaHCO_3_	0.50	1.00	P, %	0.38	0.37
Total	100.00	100.00			

^(1)^ The premix provided the following per kg of diets: Ca 1520.0 mg, P 410.0 mg, Fe 25.0 mg, I 0.90 mg, Zn 35.0 mg, Co 0.10 mg, Cu 9.00 mg, Se 0.25 mg, Mn 19.5 mg, V_E_ 15 IU, V_D3_ 1000 IU, V_A_ 3000 IU, and nicotinic acid 60.0 mg. ^(2)^ The ME was calculated as ME = sum of the effective energy of each ingredient × the additive ratio of each ingredient, while the other nutrient levels were measured.

**Table 2 animals-14-00182-t002:** The primer sequences of the target and internal reference genes.

NCBI Reference	Gene	Primer Sequence (5′-3′)	Product Size (bp)	Tm (°C)
NM_001009465.2	*IL-1β*	IL-1β F: CAGCCGTGCAGTCAGTAA IL-1β R: TGTGAGAGGAGGTGGAGAG	100 bp	57
NM_001009392.1	*IL-6*	IL-6 F: GGGTAAAGAACGCAAAGGT IL-6 R: TGACCAGAGGAGGGAATG	136 bp	55
NM_001145185.2	*SOD2*	SOD2 F: GCTTCGAGGCAAAGGGAGAT SOD2 R: AACTGATGGACGTGGAACCC	88 bp	60
NM_001009803.1	*IFN-γ*	IFN-γ F: CAGGAGCTACCGATTTCAAC IFN-γ R: AAACCCAAAAGCACACAGA	100 bp	56
NM_001024860.1	*TNF-α*	TNF-α F: ACACCATGAGCACCAAAAGC TNF-α R: AGGCACCAGCAACTTCTGGA	103 bp	60
NM_001161888.1	*CD93*	CD93 F: CTTGGGGAAGACACGGGAAA CD93 R: GAGGTAGCCTCAGGGTTGTC	89 bp	59
XM_004018462.5	*GPX1*	GPX1 F: AACGTAGCATCGCTCTGAGG GPX1 R: CAAACTGGTTGCACGGGAAG	115 bp	57
XM_027971670.2	*PAK6*	PAK6 F: CGCTGCCTGAGGATTTAATGGA PAK6 R: CTGTCAAAGGGAGTGCGGC	145 bp	61
XM_027962288.2	*VCL*	VCL F: ATGCTGTTGGGTTCCCTGTC VCL R: GACCACTTGGTAGCTTCCCG	139 bp	60
XM_042246630.1	*ITGA5*	ITGA5 F: TGCTGTGAACCAGAGTCGTC ITGA5 R: TGGGACGAGGAGAGACTGAG	159 bp	59
XM_004013602.4	*TGFB2*	TGFB2 F: TTACCCTCGGAAACTGTCTGC TGFB2 R: GGCATCAAGGTACCCACAGAA	102 bp	60
NM_001009784.3	*β-actin*	ACTB F: CATCGTCCACCGCAAAT ACTB R: GCCATGCCAATCTCATCTC	103 bp	56

**Table 3 animals-14-00182-t003:** Quality control results.

Sample	Raw Reads	Clean Reads	Error Rate (%)	Q20 (%)	Q30 (%)	GC Content (%)	Uniquely Mapped (%)
C1-1	48,466,846	47,954,082	0.0250	98.04	94.12	48.97	84.97
C1-2	56,038,432	55,523,466	0.0249	98.08	94.24	49.67	84.88
C1-3	48,483,846	48,016,986	0.0249	98.07	94.24	50.16	86.77
H2-1	50,097,052	49,600,336	0.0248	98.13	94.39	50.03	85.92
H2-2	52,449,996	51,964,618	0.0249	98.10	94.27	49.28	84.83
H2-3	48,366,524	47,724,756	0.0254	97.87	93.75	49.91	85.56
S3-1	49,668,102	49,066,542	0.0254	97.88	93.74	48.76	84.85
S3-2	50,354,892	49,889,566	0.0248	98.13	94.35	49.04	86.76
S3-3	44,943,236	44,425,012	0.0265	97.48	92.72	51.77	85.03

Raw reads: the total number of entries in the original sequencing data; clean reads: the total number of entries in the sequencing data after quality control; error rate: the average error rate of sequencing bases corresponding to the quality control data; Q20 and Q30: the percentage of bases with a sequencing quality of 99% and 99.9%, respectively, in the total number of bases; GC content: the percentage of the sum of G and C bases in the total number of bases; uniquely mapped: the number of clean reads with unique matching positions on the reference sequence.

**Table 4 animals-14-00182-t004:** Expression information for the main differentially expressed genes in the sheep liver signaling pathways.

Gene	Gene Description	FC			*p*-Value
		C1/H2	S3/H2	S3/C1	
*COL1A1*	collagen type I alpha 1 chain	0.52 **	0.35 **	0.68	0.004
*COL1A2*	collagen type I alpha 2 chain	0.53 **	0.39 **	0.74	0.004
*COL6A2*	collagen type VI alpha 2 chain	0.68 *	0.56 *	0.82	0.021
*COL6A3*	collagen type VI alpha 3 chain	0.37 **	0.34 **	0.90	0.003
*COL6A5*	collagen type VI alpha 5 chain	0.65	0.33 *	0.51	0.039
*PAK6*	p21 (RAC1)-activated kinase 6	0.24 *	0.30 *	1.25	0.034
*ARHGAP35*	Rho GTPase-activating protein 35	1.08	0.88 *	0.81 *	0.024
*PDGFB*	platelet-derived growth factor subunit B	0.74 *	0.50 **	0.67 *	0.006
*ITGA5*	integrin subunit alpha 5	0.63 *	0.99	1.59 *	0.014
*LAMC1*	laminin subunit gamma 1	0.84	0.68 *	0.81	0.038
*VCL*	vinculin	0.89	0.71 *	0.79 *	0.019

FC values indicate the fold difference in expression between the two groups. * indicates expression *p* < 0.05, and ** indicates expression *p* < 0.01.

**Table 5 animals-14-00182-t005:** q-PCR validation results.

Gene	Group			*p*-Value
	C1	H2	S3	
*IL-6*	0.62 ± 0.05 ^B^	1.00 ± 0.02 ^A^	1.04 ± 0.13 ^A^	<0.001
*INF-γ*	0.98 ± 0.13	1.00 ± 0.04	0.95 ± 0.04	0.506
*IL-1β*	1.02 ± 0.10	1.00 ± 0.03	1.08 ± 0.06	0.706
*TNF-α*	0.89 ± 0.09 ^ab^	1.00 ± 0.03 ^a^	0.78 ± 0.06 ^b^	0.018
*CD93*	0.73 ± 0.11 ^B^	1.00 ± 0.05 ^A^	0.64 ± 0.07 ^B^	0.004
*GPX1*	1.04 ± 0.04	1.00 ± 0.03	1.07 ± 0.06	0.167
*SOD2*	1.41 ± 0.02 ^A^	1.00 ± 0.05 ^C^	1.30 ± 0.02 ^B^	<0.001
*PAK6*	0.81 ± 0.05 ^b^	1.00 ± 0.04 ^a^	0.85 ± 0.11 ^b^	0.025
*ITGA5*	0.68 ± 0.03 ^B^	1.00 ± 0.04 ^A^	0.93 ± 0.13 ^A^	<0.001
*TGFB2*	0.55 ± 0.02 ^B^	1.00 ± 0.04 ^A^	0.99 ± 0.16 ^A^	0.002
*VCL*	0.98 ± 0.04 ^A^	1.00 ± 0.06 ^A^	0.82 ± 0.03 ^B^	0.001

Data are means ± SD for *n* = six lambs per group. Data marked without the same lowercase letters indicate significant differences at *p* < 0.05, and data without the same capital letters indicate extremely significant differences at *p* < 0.05.

## Data Availability

The original contributions presented in the study are included in the article/supplementary material, and further inquiries can be directed to the corresponding author. In addition, we have uploaded the RNA-Seq data to the Sequence Read Archive (SRA) in NCBI (submission: SUB13997134).

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
