# Peer review of "Effects of Sophora alopecuroides in a High-Concentrate Diet on the Liver Immunity and Antioxidant Function of Lambs According to Transcriptome Analysis"

_animals, 2024, doi:10.3390/ani14020182_

Round 1
Reviewer 1 Report
Comments and Suggestions for Authors
Effects of adding Spohora alepecuroides in a high-concentrate diet on the liver immunity and antioxidant function of lambs based on transcriptome analysis
Dear Authors,
The manuscript is interesting and describes effect high concentrate diet on liver immunity and antioxidant status in combination with Spohora alepecuroides. There are some corrections needed in case of figures, the statistical analysis (one or two-way ANOVA), and several corrections from editorial point of view.
Below I add some suggestions helpful in this process:
Line 17,18 and 21
In text is Sophora alopecuroides, must be in present in binominal nomenclature with italics: Sophora alopecuroides
Line 27-74
In abstract section justify of text must be used.
Line 81
LPS is not introduced earlier (lipopolysaccharide), maybe it is worth to add it in this place.
Line 125
Font must be changed, and in the same part of sentence,
in text of the manuscript is wrote: ‘…the test period was 15 days for the preliminary test and 60 days for the main test…’, maybe better change for ‘…The experiment was divided on 15 days of preliminary period (fulfilling of gastrointestinal tract individual diet) and 60 days of main period…’ .
Line 187
In the text is: ‘…Ishikawa [21] et al. …’ , must be Ishikawa et al. [21].
Line 264
In my opinion is better to use one-way ANOVA, because p-value and significance for diet must be determined in each, even hour after morning feeding (p-value in 2nd, 4th, 6th, 8th, 10th hour, additionally before morning feeding to confirm lack of significant differences between groups of animals can be added).
Line 266
On the Figure 1 sd is added to each line of diet variant, SEM normally is determined for all groups, similarly like p-value in each period [sd from all replications/ sqr root from number of all replications (24)].
Line 296-297
Figure 3e, p-value is 0,086, null hypothesis cannot be rejected; lack differences between diets, even if the post hoc test confirm them.
Line 732-889
References
Space between each reference must be removed.
Abbreviations of Journals needed: no. 1,2,4,…
Dots in abbreviations needed: 11, 22,.. (ie. Int. J. Biol. Macromol.)
Please check all references.
Author Response
Thank you very much for taking the time to review this manuscript. Please find the detailed responses below and the corresponding revisions/corrections highlighted/in track changes in the re-submitted files.

Reviewer 2 Report
Comments and Suggestions for Authors
The abstract is too long. According to Manuscript Preparation: ,,The abstract should be a total of about 200 words maximum''. Please short it!
You used a lot of abbreviations on your manuscript. Please add an Abbreviation List in your manuscript.
Please add the content of ether extract and crude fiber content of the two typs of diets. Also add the formula that you used to calculate the ME.
Something is not clear for me!!! The nutrients level for H2 diet was increased only with 0.67MJ/KG for ME and with 0.28% for CP, while you increased the composition for H2 with 15.9% corn and 3.20% soybean meal.
Please add supplementary materials for diets composition to proove that the H2 diet was high-concentrate diet.
What was the quantity of the each meal that animals recieved?
The Rumen pH was analysed daily? Please specify!
Have you collected blood also at the begining of the trial? For comparison of the results at day 60th..... Please add supplementary materials.
Have you measured the body weight at the begining and the end of the trial? Please add supplementary materials.
You declared that you euthanasied the animals? What about the ethics commision approval? Please add the ethics committee approval!
Author Response

(The authors gave the same response as above.)

Reviewer 3 Report
Comments and Suggestions for Authors
Review of Article: “Effects of Adding Sophora alopecuroides in a High-concentrate 2 Diet on the Liver Immunity and Antioxidant Function of Lambs 3 Based on Transcriptome Analysis”.
Please see the comments below.
Line 17,18 -> Scientific name of “Sophora alopecuroides” need to be in italics.
Line 30-31 ->isn’t explained how the groups were named, Also please add comma after 24 for more clarity.
Line 33-34 -> is this correct: C1- fed with concentrate-to-forage ratio of 50:50
H2- fed with concentrate-to-forage ratio of 70:30
S3- fed with concentrate-to-forage ratio of 70-30 + 0.1% SA
Line 34 -> there should be word respectively after a comma.
Line 67 -> what are CC and BF categories? Please explain.
Line 125 ->the text has a different format. Also, different colour
Line 127 -> same as 30-31: the groups aren’t well explained (which one is the control group? If there is any)
Line 165 -> <<then were cut into small pieces with with a scalpel into the size of about 1 m3>> “with” is written 2 times.
Line 211 -> <<Extraction and sequencing of RNA from liver tissue The total RNA of all samples was>> “The” is with capital letter but there isn’t punctuation. There should be full stop.
Line 591 -> <<ditives in the sheep diet is vatal important.>> I think that there is a spelling mistake. (Vital)
Line 653 -> <<sis(PF)>> there isn’t the space between “s” and “(“
Line 670 -> <<upstream of FAs [57].FAK is a highly>> there isn’t the space between “.” and “F“
Line 677 -> << SRC3&4(EGFR-FAK bridging protein)>> there isn’t the space between “4” and “(“
Comments on the Quality of English LanguageThe English is good there are some spelling mistakes and double words, I wrote in the above mentioned file.
Author Response

(The authors gave the same response as above.)

Round 2
Reviewer 2 Report
Comments and Suggestions for Authors
The authors made the changes recommended by the reviewer.
Author Response
Thank you very much for taking the time to review this manuscript.